# Effects of Self-Leadership on Nursing Professionalism among Nursing Students: The Mediating Effects of Positive Psychological Capital and Consciousness of Calling

**DOI:** 10.3390/healthcare12121200

**Published:** 2024-06-15

**Authors:** Jiyoung Seo, Hana Ko

**Affiliations:** 1Department of Nursing, Bucheon University, 56, Sosa-ro, Sosa-gu, Bucheon-si 14774, Republic of Korea; seojiy@bc.ac.kr; 2College of Nursing, Gachon University, 191, Hambakmoe-ro, Yeonsu-gu, Incheon 21936, Republic of Korea

**Keywords:** nursing, professionalism, self-leadership, psychological capital, consciousness of calling

## Abstract

To provide high-quality nursing care, nursing education requires the basic quality of self-leadership from professional nurses so that they can make self-directed and responsible judgments and decisions. Therefore, this study aimed to investigate relationships among self-leadership, positive psychological capital, consciousness of calling, and nursing professionalism in nursing students. A cross-sectional online survey of 202 students from two universities in South Korea was conducted between August and September 2022, using a convenience sampling method. A structured questionnaire was used to collect data. Data were analyzed using descriptive statistics, Pearson’s correlation coefficient analysis, and bootstrapping using Hayes’ PROCESS macro for mediation. A significant positive correlation was found between self-leadership, positive psychological capital, consciousness of calling, and nursing professionalism. Positive psychological capital and consciousness of calling showed an indirect mediating effect on the relationship between self-leadership and nursing professionalism. To improve nursing professionalism, programs should be developed to educate nursing students, strengthen their self-leadership skills, and increase the influence of positive psychological capital and consciousness of calling for nursing. This will ultimately contribute to improving the quality of patient care by fostering competent nursing experts.

## 1. Introduction

Nursing professionalism refers to the systematic view of nursing as the value orientation and standards of practicing nurses, reflecting the cognitive, attitudinal, and psychomotor dimensions of nursing [1]. Nursing professionalism plays a positive role in enhancing satisfaction with the major of nursing and caring behavior by allowing nursing students to clearly set career goals [2,3,4]. In addition, it plays an important role in increasing nurses’ intention to work and improve their clinical performance [5]. Therefore, it is important to ensure that nursing students, as future nurses, form the right values for nursing professionalism through the curriculum [2,6]. These values are not only influenced by individual perceptions, attitudes, personality, and motivations but also by the social perception of nursing professionals [7]. Therefore, it is necessary to confirm the factors affecting the nursing professionalism of nursing students.

In addition to the increased demand caused by the COVID-19 pandemic, the level of expectations and demands of patients have increased, requiring more expertise from nurses [8]. To provide high-quality nursing care in this scenario, nursing education requires self-leadership as a basic quality of professional nurses who can make self-directed and responsible judgments and decisions [9]. Self-leadership is the process of managing and influencing oneself through the effective use of behavioral and cognitive strategies and involves motivation through self-management, such as goal establishment, self-reinforcement, self-correction, and feedback [10]. It is the ability to be influenced by these variables and can be learned through training [11]. Studies have shown that self-leadership is positively correlated with nursing professionalism, and other personal characteristic variables, such as social support, self-efficacy, and resilience, were significantly correlated with self-leadership [11,12]. Therefore, it is important to analyze the level of self-leadership in nursing students and to understand the relationship between self-leadership and nursing professionalism. Identifying the parameters that affect this relationship will have implications, such as finding new ways to establish nursing professionalism.

Recently, the variable of positive psychological capital has been suggested as an important factor that has a positive effect on nursing professionalism [13]. Positive psychological capital refers to a positive psychological state in which individuals pursue development and is a high-level concept in which four sub-domains are integrated: self-efficacy, hope, optimism, and resilience [14]. Positive self-efficacy and hope increase responsiveness to difficulties; optimism helps maintain a positive attitude in challenging situations; and resilience helps individuals overcome work stress or burden [15]. Positive psychological capital in undergraduate nursing students was identified as a specific predictor of specialty satisfaction [16], as it enhances and stabilizes the level of their professional commitment and reduces the risk of learning burnout [17]. Further, in crisis situations, such as COVID-19, positive psychological capital has been found to be an essential factor for mediating individuals’ grit and ability to persevere, have passion, and be committed to achieving long-term goals regardless of adversity or challenge [18]. Therefore, the development of positive psychological capital in nursing students can have a positive impact not only on their academic achievement and career development but also on the process of adopting the nursing profession and the nursing care of patients. Previous research has confirmed a positive correlation between self-leadership and positive psychological capital [19] and between positive psychological capital and nursing professionalism [13]. However, research on the mediating effect of the level of positive psychological capital, a personal characteristic, in the relationship between self-leadership and nursing professionalism is still insufficient. Positive psychological capital is an individual’s strength and character and can be improved through training and learning, such as self-leadership [14]. Therefore, if the mediating effect of positive psychological capital is verified through this study, it will provide a theoretical basis for including the strengthening of positive psychological capital to develop a nursing education program for nursing students to establish nursing professionalism.

In an environment where clinical practice education for nursing students is difficult and nurses’ professional commitment is required owing to the COVID-19 pandemic, nursing professionalism and their consciousness of calling, a new variable, should be reconfirmed as an influencing factor [20,21]. Although calling has been understood as a Christian religious concept of “God calls people through their own work”, proposed by Luther and Calvin in the 16th century, it is extended to a secular concept that leads an individual to feel a sense of personal fulfillment or socially meaningful commitment [22]. A calling to nursing refers to individuals’ desire to participate in nursing practice as a means of fulfilling their purpose in life to help others with passionate intrinsic motivations [23]. The calling of nursing and nursing professionalism have been reported to be motivating and applied to clinical sites to care for patients with COVID-19 [24]. Further, nurses’ images were delivered to people through the media, affecting not only the public but also nursing students’ awareness and attitudes toward the satisfaction of nursing majors and nursing professionalism [25]. Previous studies have confirmed a significant correlation between nursing students’ consciousness of calling and nursing professionalism [20]; however, no studies have confirmed the mediating effect of consciousness of calling on the relationship between self-leadership and nursing professionalism. Self-leadership is a driving force for self-management by influencing oneself to achieve self-set goals, and it can be expected to affect the consciousness of calling by giving a purpose and meaning to nursing professionals who care for patients. Therefore, it is important to verify the mediating effect of consciousness of calling on the relationship between self-leadership and nursing professionalism.

The theoretical models that guided this study were self-regulation theory and positive organizational behavior (POB). Self-regulation theory suggests that behavior is goal-directed and feedback-controlled and that the goals underlying behavior form a hierarchy of abstractness [26]. Goal constructs bring a dynamic quality to the conceptualization of the self, and possible selves are future-oriented. In self-regulation theory, perseverance, optimism, and self-efficacy are factors related to goal pursuit that lead to positive outcomes in life [26]. Positive organizational behavior is the organizational application of positive psychology, the study and application of positively oriented human-resource strengths and psychological capacities that can be measured, developed, and effectively managed for performance improvement [27]. The criteria-meeting capacities for POB are self-efficacy, optimism, hope, and resilience and, when combined, represent what has been termed psychological capital [27]. In POB, positive psychological capital plays a mediating and coordinating role in an individual’s job-related attitudes and behaviors, which in turn, drives positive organizational change [14,28]. Based on this theoretical basis and previous research, this study attempted to present new implications for nursing education by uncovering the mediation effects of positive psychological capital and consciousness of calling, which are individual strengths, in the path to achieving the goal of improving nursing professionalism through self-leadership in nursing students.

Consequently, this study aimed to investigate the relationship among self-leadership, positive psychological capital, consciousness of calling, and nursing professionalism in nursing students. Furthermore, we aimed to identify the mediating effects of positive psychological capital and consciousness of calling on the relationship between self-leadership and nursing professionalism. This study will provide a basis for the development of a nursing-professionalism education program that is necessary to apply on an appropriate basis for nursing students.

## 2. Materials and Methods

### 2.1. Study Design

This study adopted a descriptive cross-sectional design to investigate the mediating effect of positive psychological capital and consciousness of calling on the relationship between self-leadership and nursing professionalism among nursing students.

### 2.2. Participants

Convenience sampling was used to select participants who were Korean undergraduate nursing students enrolled in two different universities located in G City. The sample size was estimated using the G*Power 3.1.9.2 program (Heinreich-Heine-Universität, Düsseldorf, Germany [Heinrich Heine University Dusseldorf, Germany]) to evaluate the effects of a regression analysis on nursing professionalism. The minimum number of participants was 178 to facilitate a statistical power of 0.95 at a significance level of 0.05 and a median effect size of 0.15, with a conservative effect size of 11 predictors. We recruited 210 participants, considering an online-survey dropout rate of approximately 20%; 202 valid completed questionnaires were used in the final analysis.

### 2.3. Measurement

General characteristics such as age, sex, college year, religion, grade in the previous semester, the motive for selecting a major, personality, and satisfaction toward a nursing major are included in our analysis. Previous studies have identified general characteristics as antecedents of nursing professionalism, as well as demographic characteristics such as education level and culture that have been shown to be influential [6,7].

#### 2.3.1. Self-Leadership

Self-leadership was measured using the Abbreviated Self-Leadership Questionnaire (ASLQ) developed by Houghton, Dawley, and DiLiello [29], and validated with the Korean population by Choi [30]. The instrument contains nine items in three subdomains (three items each): behavioral awareness and will, task motivation, and constructive cognition. Responses were provided using a 5-point Likert scale ranging from 1 (not at all accurate) to 5 (completely accurate). Higher scores indicated greater self-leadership. Cronbach’s α was 0.87 and 0.76 in Choi’s study [30] and the present study, respectively.

#### 2.3.2. Positive Psychological Capital

Positive psychological capital was measured using the Korean version of Positive Psychological Capital (K-PPC), developed by Luthans et al. [31] and validated in the Korean population by Lim [32]. The questionnaire includes 18 items across four subdomains: self-efficacy (five items), optimism (five items), hope (five items), and resilience (three items). Responses were provided using a 6-point Likert scale ranging from 1 (strongly disagree) to 6 (strongly agree). Higher scores indicated greater positive psychological capital. Cronbach’s α was 0.93 and 0.92 in Lim’s [32] study and the present study, respectively.

#### 2.3.3. Consciousness of Calling

Consciousness of calling was measured using the Korean Calling and Vocation Questionnaire (CVQ-K) developed by Dik, Eldridge, and Steger [33], and validated in the Korean population by Shim and Yoo [34]. The questionnaire includes 12 items in three subdomains (four items each): transcendent sermon, purposeful work, and prosocial orientation. Responses were provided using a 4-point Likert scale ranging from 1 (not at all true of me) to 4 (totally true of me). Higher scores indicated greater consciousness of calling. Cronbach’s α was 0.85 and 0.88 in Shim and Yoo’s study [34] and the present study, respectively.

#### 2.3.4. Nursing Professionalism

Nursing professionalism was measured using the Nursing Professional Values Scale developed by Yeun et al. [35]. The questionnaire included 29 items in five subdomains: self-concept of the profession (nine items), social awareness (eight items), professionalism of nursing (five items), the roles of nursing service (four items), and originality of nursing (three items). Responses were provided using a 5-point Likert scale ranging from 1 (not at all) to 5 (highly). Higher scores indicated higher professional nursing values. Cronbach’s α was 0.92 at the time of development by Yeun et al. [35] and 0.94 in the present study.

### 2.4. Procedure and Ethical Consideration

The data were collected using a self-reported online survey distributed among undergraduate nursing students between 6 August and 26 September 2022, after obtaining approval from the Gachon University Institutional Review Board (IRB) (approval No. 1044396-202205-HR-098-01) for the ethical protection of the participants. We posted a recruitment advertisement, including an online structured questionnaire survey link, in a group chat room by grade. The first page of the survey included the necessity and purpose of the study, data collection method, etc., and a button (“I agree”), the clicking of which would denote that the participant had consented to participate in the survey. It took approximately 10–15 min per person to complete the questionnaire. Participants were informed that they could withdraw from the survey at any time without penalty if they no longer wished to participate. All participants understood the purpose of the study and voluntarily participated in it. Upon the completion of data collection, gifts were presented to the participants who completed the questionnaire.

### 2.5. Data Analysis

Data were analyzed using the SPSS/WIN 25.0 program (IBM Corp., Armonk, NY, USA) and SPSS/WIN PROCESS macro v3.4. Frequencies, percentages, means, and standard deviations were computed to examine the participants’ general characteristics, self-leadership, positive psychological capital, consciousness of calling, and nursing professionalism. In addition, *t*-tests and an analysis of variance (ANOVA) with Scheffe’s post hoc tests were performed to examine the differences in satisfaction with the nursing major according to participants’ general characteristics. Pearson’s correlation coefficients were calculated to explore correlations between the main study variables. The mediating effects of positive psychological capital and consciousness of calling on the relationship between self-leadership and nursing professionalism were analyzed using the Hayes’ PROCESS macro model 4 [36]. The statistical significance of the indirect effect on each outcome variable was assessed by bootstrapping (5000 samples) with a 95% confidence interval. Also, this study declared a significance level of *p* < 0.05.

## 3. Results

### 3.1. Participants’ General Characteristics

The mean age of the participants was 22.76 years. There were 170 female students (84.2%) and 32 male students (15.8%), of which 75 (37.1%), 60 (29.7%), and 67 (33.2%) were second-, third-, and fourth-year students, respectively. In terms of grades in the previous semester, the highest proportion had a range of 3.5–3.9 (43.6%, n = 88) and a slightly lower proportion had a grade of ≥4.0 (37.1%, n = 75). Regarding the motivation behind selecting the major of nursing, the proportion of students (36.1%, n = 73) who reported aptitude or interest was the largest, followed by those who reported employment after graduation (30.2%, n = 61). With respect to personality, the proportion of students with moderate personality (38.2%, n = 77) was the highest. Most of the participants (166, 82.2%) answered that they were more than satisfied with the nursing major. Of these, 117 (57.9%) were satisfied, and 49 (24.3%) were very satisfied (Table 1).

### 3.2. Nursing Professionalism Based on Participants’ General Characteristics

Upon analysis of the differences in nursing professionalism according to the participants’ general characteristics, the motive for selecting the nursing major (F = 4.513, *p* = 0.002) and satisfaction with the nursing major (F = 10.767, *p* < 0.001) showed significant differences. In other words, students who answered that the motivation behind selecting the major was aptitude or interest had a higher level of nursing professionalism (4.08 ± 0.43) compared to those whose motivation was college-entry examination score (3.61 ± 0.68). Furthermore, the students who were very satisfied with their nursing major had higher levels of nursing professionalism (4.19 ± 0.43) than those who were satisfied (3.90 ± 0.50) or less than moderately satisfied (3.71 ± 0.51) (Table 1).

### 3.3. Scores of the Main Variables

The participants’ mean self-leadership score was 3.92 out of 5. Their mean positive psychological capital, consciousness of calling, and nursing professionalism scores were 4.46 out of 6, 2.76 out of 4, and 3.94 out of 5, respectively (Table 2).

### 3.4. Correlation among Variables

Self-leadership was significantly positively correlated with positive psychological capital (r = 0.529, *p* < 0.001), consciousness of calling (r = 0.392, *p* < 0.001), and nursing professionalism (r = 0.463, *p* < 0.001). Statistically significant positive correlations were found between nursing professionalism and positive psychological capital (r = 0.454, *p* < 0.001) and consciousness of calling (r = 0.452, *p* < 0.001) (Table 3).

### 3.5. Mediating Effects of Positive Psychological Capital and Consciousness of Calling

To examine the mediating effect of positive psychological capital and consciousness of calling on the relationship between self-leadership and nursing professionalism, significance tests were conducted for each pathway after controlling for the motive for selecting a major and satisfaction toward the nursing major, which were expected to influence nursing professionalism among the general characteristics (Table 4, Figure 1 and Figure 2).

The results of the direct-effect test indicated that self-leadership was positively related to positive psychological capital (B = 0.562, *p* < 0.001) and consciousness of calling (B = 0.342, *p* < 0.001). When analyzed with self-leadership as the predictor and positive psychological capital and consciousness of calling as the dependent variable, respectively, higher self-leadership was associated with higher positive psychological capital and consciousness of calling. When analyzing self-leadership and positive psychological capital as predictors and nursing professionalism as the dependent variable, self-leadership (B = 0.289, *p* < 0.001) and positive psychological capital (B = 0.200, *p* < 0.001) had a, respectively, significant direct effect on nursing professionalism. When analyzing self-leadership and consciousness of calling as predictors and nursing professionalism as the dependent variable, self-leadership (B = 0.313, *p* < 0.001) and consciousness of calling (B = 0.258, *p* < 0.001) had a, respectively, significant direct effect on nursing professionalism.

The mediating effect of self-leadership on nursing professionalism through positive psychological capital (B = 0.112, 95% BC bootstrap CI [0.019, 0.232]) and consciousness of calling (B = 0.088, 95% BC bootstrap CI [0.033, 0.161]) was, respectively, significant because the 95% bootstrap confidence interval did not include zero.

## 4. Discussion

This research was conducted to identify the mediating effects of positive psychological capital and consciousness of calling in the relationship between self-leadership and nursing professionalism among nursing students, based on the theoretical basis of self-regulation theory and POB. Through this, the research aimed to provide new directions and ideas for developing nursing education to empower nursing professionalism among nursing students.

In this study, we provide empirical evidence for the antecedents of nursing professionalism. The result of this study suggests that the self-regulation theory and POB may provide an effective framework for exploring what process leads to nursing students’ nursing professionalism. Nurses, who practice caring for patients, have the professional capacity to positively accept and endure diverse and complex clinical situations, such as the COVID-19 pandemic [13]. As for nursing students, college is an important stage in developing nursing professionalism. Students with higher nursing professionalism will be able to contribute more to a nursing career that is more stable and more professionally satisfying [2,3,4,5]. More theoretical and practical insights are needed in this area in the future to explore the professionalism of nursing students.

Nursing professionalism varies according to the participants’ motivation for choosing and satisfaction with the nursing major. These results are similar to those of a previous study [20], which showed that the nursing professionalism of the group who chose nursing as a major according to school grades, parents’ recommendations, or ease of employment after graduation was lower than that of those who chose it because of their aptitude and interest. These factors were found to be components of the personal characteristics of the micro-antecedents of nursing professionalism in a conceptual analysis study [7] and were found to be different in this study. Therefore, to increase nursing professionalism, a deep understanding and concern about career exploration and the nursing major are required before entering a nursing college. In addition, it is necessary to find ways to improve nursing students’ satisfaction with their major while attending nursing colleges.

This study found a positive correlation between self-leadership, positive psychological capital, consciousness of calling, and nursing professionalism. Nursing students were taught to focus on helping and caring for others. They also obtained good professional identification and professional values because of a favorable social image of nurses [37]. The recent COVID-19 pandemic has highlighted these variables [13,38], so it is important to understand whether they remain positively correlated post-COVID.

Finally, the mediating effects of positive psychological capital and consciousness of calling on the relationship between self-leadership and nursing professionalism were identified. The effects of self-leadership on nursing professionalism were predicted after controlling for the motivation for selecting a major and the satisfaction of a nursing major, which were expected to influence nursing professionalism among the general characteristics. These findings imply that, although the level of self-leadership among nursing students affects their nursing professionalism, higher levels of positive psychological capital and consciousness of calling can further strengthen it. Nursing students’ high positive psychological capital enhances their learning commitment to their major and caring efficacy for patients [19,39]. Furthermore, positive psychological capital can play a good mediating role in strengthening the endurance and grit of nursing students to grow through various crises in clinical fields after graduation [18]. In Chaleoykitti and Thaiudom’s research [40], nurses who participated in a positive psychological capital program that included improvement in self-efficacy, hope, optimism, resilience, and hardiness had a higher intention of retention than nurses who did not participate. Positive psychological capital influenced the nurses’ intention to care for patients with COVID-19 [13]. Therefore, nursing educators should pay attention to raising the positive mental state of nursing students to the best. Additionally, it is necessary to research and develop a positive psychological capital promotion program that helps nursing students improve their nursing professionalism after graduation by enabling them to optimistically perceive crises, such as COVID-19, and overcome difficult and complex situations with hope and resilience. Nursing students’ consciousness of calling enhances their satisfaction with the nursing major and improves their performance in the nursing profession [20,25,41]. In addition, their consciousness of calling increases their intention to stay in clinical fields and provide care to patients with a sense of mission as nursing professionals, especially at dangerous clinical sites of COVID-19, where their health and life are threatened [21]. Therefore, to establish the right perception of nursing professionalism, nursing educators’ efforts are needed to increase the consciousness of calling that nursing students and nurses should contribute to society by fully recognizing the meaning of work and the value of nursing care through the curriculum and clinical field. Recent meta-studies [42] have shown that online self-leadership education programs are also effective in improving self-leadership skills, and based on a study that applied an image-making program [43] to students, which emphasized their beliefs and values as nurses, they were effective at improving nursing professionalism. There is a need for interventions that enhance positive psychological capital and consciousness of calling by providing examples related to nursing, rather than just general content using online, including video, live, and simulation content.

### Limitations

This study had several limitations. First, convenience sampling was used to recruit nursing students from only two colleges in two metropolitan cities in South Korea, which limits the generalizability of the results to all nursing students in other countries. Therefore, further investigation with a larger sample is indicated to further validate these findings. Second, this cross-sectional study used self-report questionnaires to collect data, which could be associated with a potential risk of response distortion. Therefore, future research should apply systematic research methods to reveal causal relationships. Despite these limitations, the significance of this study is that it has confirmed for the first time the mediating effect of positive psychological capital and consciousness of the calling level of Korean nursing students regarding nursing professionalism.

## 5. Conclusions

This study demonstrated that positive psychological capital and consciousness of calling showed indirect mediating effects on the relationship between self-leadership and nursing professionalism among nursing students. Therefore, to improve the nursing professionalism of nursing students, it is necessary to educate them by developing programs that can strengthen self-leadership and increase the influence of positive psychological capital and consciousness of calling for nursing. This will ultimately contribute to improving the quality of patient care by fostering competent nursing experts.

## Figures and Tables

**Figure 1 healthcare-12-01200-f001:**
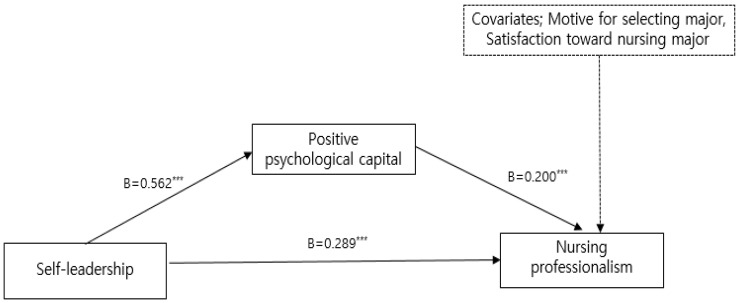
Mediating effects of self-leadership on nursing professionalism through positive psychological capital. Adjusted for motive for selecting major, satisfaction toward nursing major. Total effect = 0.401; direct effect = 0.289; indirect effect = 0.112 (95% BC bootstrap CI = 0.019–0.232), *** *p* < 0.001.

**Figure 2 healthcare-12-01200-f002:**
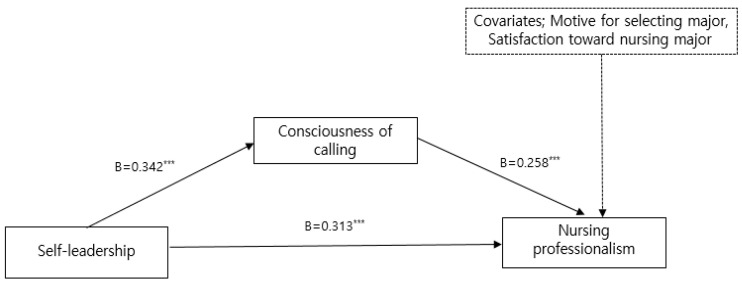
Mediating effects of self-leadership on nursing professionalism through consciousness of calling on the relationship. Adjusted for motive for selecting major, satisfaction toward nursing major. Total effect = 0.401; direct effect = 0.313; indirect effect = 0.088 (95% BC bootstrap CI = 0.033–0.161), *** *p* < 0.001.

**Table 1 healthcare-12-01200-t001:** General characteristics and differences in nursing professionalism among participants (*N* = 202).

Variables	Categories	Mean ± SD	Nursing Professionalism	
		or n (%)	Item Mean ± SD	t/F (*p*)
Age (years)		22.76 ± 3.71		0.834 (0.436)
	≤24	165 (81.7)	3.95 ± 0.53	
	25–29	24 (11.9)	3.92 ± 0.42	
	≥30	13 (6.4)	3.76 ± 0.48	
Sex	Male	32 (15.8)	3.88 ± 0.62	−0.632 (0.528)
	Female	170 (84.2)	3.95 ± 0.50	
College year	Second	75 (37.1)	3.95 ± 0.52	2.708 (0.069)
	Third	60 (29.7)	4.04 ± 0.59	
	Fourth	67 (33.2)	3.83 ± 0.42	
Religion	Yes	67 (33.2)	3.99 ± 0.52	0.987 (0.325)
	No	135 (66.8)	3.91 ± 0.52	
Grade previous semester	≥4.0	75 (37.1)	4.02 ± 0.45	1.613 (0.202)
	3.5–3.9	88 (43.6)	3.88 ± 0.58	
	<3.5	39 (19.3)	3.90 ± 0.49	
Motive for selecting major	Aptitude/interests ^a^	73 (36.1)	4.08 ± 0.43	4.513 (0.002) a > b
	College-entry examination score ^b^	19 (9.4)	3.61 ± 0.68	
	Employment after graduation ^c^	61 (30.2)	3.82 ± 0.50	
	Parent’s and teacher’s recommendation ^d^	28 (13.9)	4.02 ± 0.58	
	Service/dedication ^e^	21 (10.4)	3.98 ± 0.41	
Personality	Extrovert	54 (26.7)	3.93 ± 0.48	1.004 (0.368)
	Introvert	71 (35.1)	3.88 ± 0.55	
	Ambivert	77 (38.2)	4.00 ± 0.51	
Satisfaction toward nursing major	Very satisfied ^a^	49 (24.3)	4.19 ± 0.43	10.767 (<0.001) a > b,c
	Satisfied ^b^	117 (57.9)	3.90 ± 0.50	
	Less than moderately satisfied ^c^	36 (17.8)	3.71 ± 0.54	

Notes. SD, standard deviation; a–e means the result of post-hoc test of Scheffe.

**Table 2 healthcare-12-01200-t002:** Score of the main variables (*N* = 202).

Variables	Range	Item Mean ± SD	Min–Max
Self-leadership	1–5	3.92 ± 0.52	2.44–5.00
Positive psychological capital	1–6	4.46 ± 0.66	1.61–5.83
Consciousness of calling	1–4	2.76 ± 0.59	1.17–4.00
Nursing professionalism	1–5	3.94 ± 0.52	1.34–5.00

Notes. SD, standard deviation.

**Table 3 healthcare-12-01200-t003:** Correlation among variables (*N* = 202).

Variables	SL	PPC	CC	NP
	r (*p*)			
SL	1			
PPC	0.529 (<0.001)	1		
CC	0.392 (<0.001)	0.485 (<0.001)	1	
NP	0.463 (<0.001)	0.454 (<0.001)	0.452 (<0.001)	1

Note. SL, self-leadership; PPC, positive psychological capital; CC, consciousness of calling; NP, nursing professionalism.

**Table 4 healthcare-12-01200-t004:** Mediating effects of positive psychological capital and consciousness of calling on the relationship between self-leadership and nursing professionalism (*N* = 202).

Outcome	Predictors	B	SE	t	*p*	LLCI	ULCI
Positive psychological capital	Constant	2.760	0.373	7.393	<0.001	2.024	3.497
	Self-leadership	0.562	0.078	7.238	<0.001	0.409	0.716
	R^2^ = 0.343, F = 34.483, *p* < 0.001						
Nursing professionalism	Constant	2.109	0.345	6.118	<0.001	1.429	2.789
	Self-leadership	0.289	0.071	4.048	<0.001	0.148	0.043
	Positive psychological capital	0.200	0.058	3.438	0.001	0.085	0.314
	R^2^ = 0.286, F = 19.685, *p* < 0.001						
Consciousness of calling	Constant	1.881	0.362	5.200	<0.001	1.167	2.594
	Self-leadership	0.342	0.075	4.541	<0.001	1.193	0.490
	R^2^ = 0.088, F = 6.389, *p* < 0.001						
Nursing professionalism	Constant	2.175	0.320	6.803	<0.001	1.544	2.806
	Self-leadership	0.313	0.066	4.775	<0.001	0.184	0.443
	Consciousness of calling	0.258	0.059	4.377	<0.001	0.142	0.374
	R^2^ = 0.310, F = 22.106, *p* < 0.001						

Note. LLCI = low-limit confidence interval; ULCI = upper-limit confidence interval.

## Data Availability

The original contributions presented in the study are included in the article; further inquiries can be directed to the corresponding author.

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
