# Peer review of "Effects of Self-Leadership on Nursing Professionalism among Nursing Students: The Mediating Effects of Positive Psychological Capital and Consciousness of Calling"

_healthcare, 2024, doi:10.3390/healthcare12121200_

Round 1
Reviewer 1 Report
Comments and Suggestions for Authors
The paper presents a cross-sectional research design based on a survey questionnaire to examine the level of nursing professionalism in relation to self-leadership, positive psychological capital and consciousness of calling in nursing students.The study intends to adds to literature information about some significant factors that impact nursing professionalism, with the aim of substantiating effective educational programs for the professional development of nurses. The title of the article is relevant to the content of the manuscript and expresses the analyses performed. The abstract clearly summarizes the aim, method and the main results of the study. Overall, I find the introduction relevant to grounding the empirical study and the references cited adequate and recent. The authors define the main concept of the study et describe previous empirical research, focusing especially on correlational analyses. In general, the methods are suitable for exploring the relationship between variables.
However, to increase the manuscript's clarity, precision and scientific power, the following aspects should be improved.
Introduction
1. The authors defined the main concepts of the study, but did not succinctly describe the theoretical framework in which each of them was developed. Is there a theory that developed self-leadership, positive psychological capital and professionalism concept, in the field of positive psychology (e.g., positive organizational behavior), human resources or others? This minor addition would frame the topic of the article (nursing professionalism) into the broader scientific field of human resources psychology.
2. A careful reading of the text is necessary, so that the expression is as clear as possible. Perhaps a review of the text by an English professional is needed. There are unclear or annoyingly repetitive sentences that make reading difficult. For example:
· Please review this sentence in the abstract: „A cross-sectional online survey of two universities was conducted in South Korea, using a convenience sampling method between August and September 2022”
· Please review this sentence - Lines 53-55. „Self-leadership was positively correlated with nursing professionalism, and other personal characteristic variables, such as social support, self-efficacy, and resilience, were significantly correlated with self-leadership”
· Lines 57-58. „Therefore, it is important to understand the level of self-leadership and nursing professionalism in future nursing professionals.” – In my opinion, „to understand the level” is not the most appropriate expression. „…to analyse the level of…. and to understand the relationship between….”.
· The sentence below is difficult to read and somewhat repetitive. A little rewording might make the sentence clearer. Lines 76-79. „Since positive psychological capital, such as self-leadership, can be improved through training and learning [15], it is meaningful to use it as a fundamental concept for developing educational programs of nursing curriculum to improve self-leadership and positive psychological capital in nursing students to establish nursing professionalism.” - There are two different concepts here. Will educational programs include only the first or also the second?
3. The authors explained the mediating effect of consciousness of calling, based on the literature, but not that of positive psychological capital. Why does self-leadership exert effects on nursing professionalism through psychological capital?
Materials and Methods
1. In section 2. Materials and Methods - it should be explained why certain general characteristics were included in the analysis. Why were motive for selecting major and satisfaction toward nursing major taken into account? Probably to control these variables. The results show differences in professionalism depending on these two general characteristics. Shouldn't they be entered into the regression model to be controlled for? However, an explanation of why these variables were analysed is needed.
2. I think section 3.3 Scores of the variables could be called 3.3 Scores of the main variables, since you have previously described other variables.
3. Abbreviated Self-Leadership Questionnaire (ASLQ) includes three subdomains: behavioral awareness and will, task motivation, and constructive cognition. But in the introduction, this operational definition of self-leadership does not appear, but others do. It would seem that in the theoretical foundation the defined concept has other dimensions than the empirically investigated one. I recommend adding this operational definition to the introduction as well, for consistency.
Results
The results were correctly presented in the tables and figure and in accordance with the analysis methods.
Discussion
Some claims have no statistical support. Therefore, the authors should exercise caution in interpretation. I think that the discussion should be slightly rewritten to avoid some speculation.
For exemple:
1. The authors interpret the scores obtained for the predictor variables compared to the scores obtained in other previous studies. Even if a difference is observed between the mean scores, we cannot draw conclusions about the differences because we do not have a statistical test to verify this. Is the mean score of 4.46 significantly higher than the score of x from the other research? (Lines 302-303: In this study, the mean positive psychological capital score was 4.46, which is a higher score than the score reported for nursing students before COVID-19 in previous studies). The same for the other interpretations of the scores. In my opinion, these comparisons are not supported.
2. In my opinion, the discussions must focus on the research objectives - the interpretation of the relationships between the variables, by reference to previous research, what the authors have already done. But even here caution is needed in interpretation. (Lines 296-297. These results support previous findings [13,14,17,20]. This means that self-leadership in nursing students is more positively correlated with positive psychological capital, consciousness of calling, nursing professionalism. How did correlation coefficients from different studies compare?
Author Response
We greatly appreciate your thoughtful comments that helped improve the manuscript.
We reflected your suggestion in the manuscript.
Please see the attachment.
Please find the revised text in red colored highlights.
I look forward to your reply regarding our manuscript.

Reviewer 2 Report
Comments and Suggestions for Authors
Abstract:
Line 13 - change "In this backdrop" to "Therefore,"
Line 17 change to " A significant positive correlation was found between self-leadership, positive psychological capital....
Introduction:
Line 40 - I am not sure this is true? Were nurses treated as assistants? I think that the public perception of nursing and media portrayal of nurses is a long and complex issue that cannot be summarized as something that occurred just during COVID 19. I would suggest re-working this section to emphasize professionalism among nursing students and removing the media portrayal during COVID 19. In addition, the complexity of patient care has been changing for many years. COVID 19 has impacted this but again is a complex issue.
Line 50-51 combine these 2 sentences to read "Self-leadership involves motivation through self-management, such as goal establishment, self-reinforcement, self-correction and feedback."
Line 53 - change the location of this sentence, move the sentence Self leadership is the ability to be influenced by these variables...to right after the sentence from line 50. Then add the sentence from line 53 but change it to "Studies have shown self-leadership is positively correlated..." Then conclude with "therefore it is important to understand....
Suggest adding a specific "Literature Review" section that discusses previous work around self-leadership and the variables that are measured in this study - possibly evidence that was gathered pre covid and post covid to show there are differences.
Discussion
Make the first paragraph longer than 2 sentences or combine it with the next paragraph.
Line 288 - rewrite sentence to say "Considering that most of the classes that were taught online during the COVID-19 pandemic continued as online courses" if that is what is meant.
Author Response

(The authors gave the same response as above.)

Reviewer 3 Report
Comments and Suggestions for Authors
I have reviewed the study “Effects of Self-leadership on Nursing Professionalism among Nursing Students: The Mediating Effects of Psychological Capital and Consciousness of Calling Study”. The topic is relevant and adds empirical evidence to nursing literature. Nevertheless, some suggestions are made.
Abstract:
I suggest that briefly describe the sample in the abstract (at least number of subjects).
Introduction and theoretical framework
Relationships are empirically well argued, but I wonder which theoretical basis are under these relationships. I am thinking self- regulation theory (Carver & Scheier, 1998). You could review the theory and add a theoretical view in order to improve the draft.
Materials and Method
Participants
I suggest that describes the sample in this section in the place of results, and add the method to calculate the sample in data analysis.
Additionally, I do not know what means “moderate personality”, and either how it is measured, could you explained.
Data analysis
Results are well-done from my point of view, but data analysis is old-fashioned. At first place, to try to test if there is common method of variance you need to calculate a measurement model. Additionally, if you do a regression model, you could not know which are indirect effects. So, here it goes three options:
- A SEM model (you could avoid measurement model)
- A path analysis.
- In Spss, you could use process macro to calculate indirect effects.
Although Sobel test showed that the mediating effect is significant, nowadays, it is possible to report more info about indirect effects.
Results
Results are well done (in the old-fashioned way). The only thing that I suggest is that you follow the APA style for the diagram. Anyway, my main suggestion is to change the analysis to improve the articel.
Discussion
I suggested that you discuss your results, in deep. You describe your descriptive results in deep, but your model is only related with other studies. I wonder what the theoretical contributions and practical are.
You could add to your limitations: common method of variance and social desirability bias, that it seems that you describe above.
At last, I propose that add more practical implications of this study.
Carver, C.S. and Scheier, M.F. (1998), On the Self-Regulation of Behavior, Cambridge University Press, New York, NY.
Author Response

(The authors gave the same response as above.)

Round 2
Reviewer 3 Report
Comments and Suggestions for Authors
Dear authors
I congratulale you for the improvement in the theoretical development and discussion. The fact that they did not change methodology, I recommend they will read (Mackinnon, Lockwood, Hoffman, West & Sheets, 2002) and consider if Sobel test and Baron and Kenny are the most suitable.
MacKinnon, D. P., Lockwood, C. M., Hoffman, J. M., West, S. G., & Sheets, V. (2002).
A comparison of methods to test mediation and other intervening variables
effects. Psychological Methods, 7, 83e104.
Author Response
We thank you and the reviewers for your thoughtful suggestions and insights.
The manuscript has benefited from these insightful suggestions. I look forward to working with you and the reviewers
to move this manuscript closer to publication in the Healthcare.
The manuscript has been rechecked and the necessary changes have been made in accordance with the reviewers’ suggestions. The responses to all comments have been prepared and attached herewith.
Thank you for your consideration. I look forward to hearing from you.
Please find the revised text in blue colored highlights.
Please see the attachment.
I look forward to your reply regarding our manuscript.
